# What do row and column marginals reveal about your dataset?

**Behzad Golshan**
Boston University
behzad@cs.bu.edu

**John W. Byers**
Boston University
byers@cs.bu.edu

**Evimaria Terzi**
Boston University
evimaria@cs.bu.edu

## Abstract

Numerous datasets ranging from group memberships within social networks to purchase histories on e-commerce sites are represented by binary matrices. While this data is often either proprietary or sensitive, aggregated data, notably row and column marginals, is often viewed as much less sensitive, and may be furnished for analysis. Here, we investigate how these data can be exploited to make inferences about the underlying matrix $H$. Instead of assuming a generative model for $H$, we view the input marginals as constraints on the dataspace of possible realizations of $H$ and compute the probability density function of particular entries $H(i, j)$ of interest. We do this for all the cells of $H$ simultaneously, without generating realizations, but rather via *implicitly sampling* the datasets that satisfy the input marginals. The end result is an efficient algorithm with asymptotic running time the same as that required by standard sampling techniques to generate a single dataset from the same dataspace. Our experimental evaluation demonstrates the efficiency and the efficacy of our framework in multiple settings.

## 1 Introduction

Online marketplaces such as Walmart, Netflix, and Amazon store information about their customers and the products they purchase in binary matrices. Likewise, information about the groups that social network users participate in, the "Likes" they make, and the other users they "follow" can also be represented using large binary matrices. In all these domains, the underlying data (i.e., the binary matrix itself) is often viewed as proprietary or as sensitive information. However, the data owners may view certain aggregates as much less sensitive. Examples include revealing the popularity of a set of products by reporting total purchases, revealing the popularity of a group by reporting the size of its membership, or specifying the in- and out-degree distributions across all users.

Here, we tackle the following question: "*Given the row and column marginals of a hidden binary matrix H, what can one infer about H?*".

Optimization-based methods for addressing this question, e.g., least squares or maximum likelihood, assume a generative model for the hidden matrix and output an estimate $\widehat{H}$ of $H$. However, this estimate gives little guidance as to the structure of the feasible solution space; for example, $\widehat{H}$ may be one of many solutions that achieve the same value of the objective function. Moreover, these methods provide little insight about the estimates of particular entries $H(i, j)$.

In this paper, we do not make any assumptions about the generative process of $H$. Rather, we approach the above question by viewing the row and column marginals as *constraints* that induce a dataspace $\mathcal{X}$, defined by the set of all matrices satisfying the input constraints. Then, we *explore* this dataspace not by estimating $H$ at large, but rather by computing the *entry-wise PDF* $\mathbb{P}(i, j)$, for every entry $(i, j)$, where we define $\mathbb{P}(i, j)$ to be the probability that cell $(i, j)$ takes on value 1 in the datasets in $\mathcal{X}$. From the application point of view, the value of $\mathbb{P}(i, j)$ can provide the data analyst

with valuable insight: for example, values close to 1 (respectively 0) give high confidence to the analyst that $H(i, j) = 1$ (respectively $H(i, j) = 0$).

A natural way to compute entry-wise PDFs is by sampling datasets from the induced dataspace $\mathcal{X}$. However, this dataspace can be vast, and existing techniques for sampling binary tables with fixed marginals [6, 9] fail to scale. In this paper, we propose a new efficient algorithm for computing entry-wise PDFs by *implicitly sampling* the dataspace $\mathcal{X}$. Our technique can compute the entry-wise PDFs of all entries in running time the same as that required for state-of-the art sampling techniques to generate just a single sample from $\mathcal{X}$. Our experimental evaluation demonstrates the efficiency and the efficacy of our technique for both synthetic and real-world datasets.

**Related work:** To the best of our knowledge, we are the first to introduce the notion of entry-wise PDFs for binary matrices and to develop implicit sampling techniques for computing them efficiently. However, our work is related to the problem of sampling from the space of binary matrices with fixed marginals, studied extensively in many domains [2, 6, 7, 9, 21], primarily due to its applications in statistical significance testing [14, 17, 20]. Existing sampling techniques all rely on explicitly sampling the underlying dataspaces (either using MCMC or importance sampling) and while these methods can be used to compute entry-wise PDFs, they are inefficient for large datasets.

Other related studies focus on identifying interesting patterns in binary data given itemset frequencies or other statistics [3, 15]. These works either assume a generative model for the data or build the maximum entropy distribution that approximates the observed statistics; whereas our approach makes no such assumptions and focuses only on exact solutions. Finally, considerable work has focused on counting binary matrices with fixed marginals [1, 8, 10, 23]. One can compute the entry-wise PDFs using these results, albeit in exponential time.

## 2   Dataspace Exploration

Throughout the rest of the discussion we will assume an $n \times m$ 0–1 matrix $H$, which is *hidden*. The input to our problem consists of the dimensionality of $H$ and its row and column marginals provided as a pair of vectors $\langle \mathbf{r}, \mathbf{c} \rangle$. That is, $\mathbf{r}$ and $\mathbf{c}$ are $n$-dimensional and $m$-dimensional integer vectors respectively; entry $\mathbf{r}(i)$ stores the number of 1s in the $i$th row of $H$, and similarly for $\mathbf{c}(i)$. In this paper we address the following high-level problem:

**Problem 1.** *Given $\langle \mathbf{r}, \mathbf{c} \rangle$, what can we infer about $H$? More specifically, can we reason about entries $H(i, j)$ without access to $H$ but only its row and column marginals?*

Clearly, there are many possible ways of formalizing the above problem into a concrete problem definition. In Section 2.1 we describe some mainstream formulations and discuss their drawbacks. In Section 2.2 we introduce our *dataspace exploration framework* that overcomes these drawbacks.

### 2.1   Optimization–based approaches

Standard optimization-based approaches for Problem 1 usually assume a generative model for $H$, and estimate it by computing $\widehat{H}$, the best estimate of $H$ optimizing a specific objective function (e.g., likelihood, squared error). Instantiations of these methods for our setting are described next.

**Maximum-likelihood (ML):** The ML approach assumes that the hidden matrix $H$ is generated by a model that only depends on the observed marginals. Then, the goal is to find the model parameters that provide an estimate $\widehat{H}$ of $H$ while maximizing the likelihood of the observed row and column marginals. A natural choice of such a model for our setting is the *Rasch model* [4, 19], where the probability of entry $H(i, j)$ taking on value 1 is given by:

$$\Pr\left[H(i, j) = 1\right] = \frac{e^{\alpha_i - \beta_j}}{1 + e^{\alpha_i - \beta_j}}.$$

The maximum-likelihood estimates of the $(n + m)$ parameters $\alpha_i$ and $\beta_j$ of this model can be computed in polynomial time [4, 19]. For the rest of the discussion, we will use the term Rasch to refer to the experimental method that computes an estimate of $H$ using this ML technique.

**Least-squares (LS):** One can view the task of estimating $H(i, j)$ from the input aggregates as solving a linear system defined by equations: $\mathbf{r} = H \times \vec{\mathbf{1}}$ and $\mathbf{c} = H^T \times \vec{\mathbf{1}}$ Unfortunately, such

a system of equations is typically highly under-determined and standard LS methods approach it as a regression problem that asks for an estimate $\widehat{H}$ of $H$ to minimize $F(\widehat{H}) = ||(\widehat{H} \times \vec{1}) - \mathbf{r}||_F + ||(\widehat{H}^T \times \vec{1}) - \mathbf{c}||_F$, where $|| \,\, ||_F$ is the Frobenius norm [13]. Even when the entries of $\widehat{H}$ are restricted to be in $[0, 1]$, it is not guaranteed that the above regression-based formulation will output a reasonable estimate of $H$. For example, all tables with row and column marginals $\mathbf{r}$ and $\mathbf{c}$ are 0-error solutions; yet there may be exponentially many such matrices. Alternatively, one can incorporate a "regularization" factor $J()$ and search for $\widehat{H}$ that minimizes $F(\widehat{H}) + J(\widehat{H})$. For the rest of this paper, we consider this latter approach with $J(\widehat{H}) = (\widehat{H}(i, j) - \overline{h})^2$, where $\overline{h}$ is the average value over all entries of $\widehat{H}$. We refer to this approach as the LS method.

Although one can solve (any of) the above estimation problems via standard optimization techniques, the output of such methods is a *holistic* estimate of $H$ that gives no insight on how many solutions $\widehat{H}$ with the same value of the objective function exist or the confidence in the value of every cell. Moreover, these techniques are based on assumptions about the generative model of the hidden data. While these assumptions may be plausible, they may not be valid in real data.

### 2.2 The dataspace exploration framework

To overcome the drawbacks of the optimization-based methods, we now introduce our *dataspace exploration framework*, which does not make any structural assumptions about $H$ and considers the set of *all possible datasets* that are consistent with input row and column marginals $\langle \mathbf{r}, \mathbf{c} \rangle$. We call the set of such datasets the $\langle \mathbf{r}, \mathbf{c} \rangle$-*dataspace*, denoted by $\mathcal{X}_{\langle \mathbf{r}, \mathbf{c} \rangle}$, or $\mathcal{X}$ for short.

We translate the high-level Problem 1 into the following question: *Given $\langle \mathbf{r}, \mathbf{c} \rangle$, what is the probability that the entry $H(i, j)$ of the hidden dataset takes on value 1?* That is, for each entry $H(i, j)$ we are interested in computing the quantity:

$$\mathbb{P}(i, j) = \sum_{H' \in \mathcal{X}} \Pr(H')\Pr\left[H'(i, j) = 1\right]. \tag{1}$$

Here, $\Pr(H')$ encodes the prior probability distribution over all hidden matrices in $\mathcal{X}$. For a uniform prior, $\mathbb{P}(i, j)$ encodes the fraction of matrices in $\mathcal{X}$ that have 1 in position $(i, j)$. Clearly, for binary matrices, $\mathbb{P}(i, j)$ determines the PDF of the values that appear in cell $(i, j)$. Thus, we call $\mathbb{P}(i, j)$ the *entry-wise PDF* of entry $(i, j)$, and $\mathbb{P}$ the *PDF matrix*. If $\mathbb{P}(i, j)$ is very close to 1 (or 0), then over all possible instantiations of $H$, the entry $(i, j)$ is, with high confidence, 1 (or 0). On the other hand, $\mathbb{P}(i, j) \simeq 0.5$ signals that in the absence of additional information, a high-confidence prediction of entry $H(i, j)$ cannot be made.

Next, we discuss algorithms for estimating entry-wise PDFs efficiently. Throughout the rest of the discussion we will adopt Matlab notation for matrices: for any matrix $M$, we will use $M(i, :)$ to refer to the $i$-th row, and $M(:, j)$ to refer to the $j$-th column of $M$.

## 3 Basic Techniques

First, we review some basic facts and observations about $\langle \mathbf{r}, \mathbf{c} \rangle$ and the dataspace $\mathcal{X}_{\langle \mathbf{r}, \mathbf{c} \rangle}$.

**Validity of marginals:** Given $\langle \mathbf{r}, \mathbf{c} \rangle$ we can decide in polynomial time whether $|\mathcal{X}_{\langle \mathbf{r}, \mathbf{c} \rangle}| > 0$ either by verifying the Gale-Ryser condition [5] or by constructing a binary matrix with the input row and column marginals, as proposed by Erdös, Gallai, and Hakimi [18, 11]. The second option has the comparative advantage that if $|\mathcal{X}_{\langle \mathbf{r}, \mathbf{c} \rangle}| > 0$, then it also outputs a binary matrix from $\mathcal{X}_{\langle \mathbf{r}, \mathbf{c} \rangle}$.

**Nested matrices:** Building upon existing results [18, 11, 16, 24], we have the following:

**Lemma 1.** *Given the row and column marginals of a binary matrix as $\langle \mathbf{r}, \mathbf{c} \rangle$ we can decide in polynomial time whether $|\mathcal{X}_{\langle \mathbf{r}, \mathbf{c} \rangle}| = 1$ and if so, completely recover the hidden matrix $H$.*

The binary matrices that can be uniquely recovered are called *nested matrices* and have the property that in their representation as bipartite graphs they do not have any *switch boxes* [16]: a pair of edges $(u, v)$ and $(u', v')$ for which neither $(u, v')$ nor $(u', v)$ exist.

**Explicit sampling:** One way of approximating $\mathbb{P}(i, j)$ for large dataspaces is to first obtain a uniform sample of $N$ binary matrices from $\mathcal{X} : X_1, \ldots, X_N$ and for each $(i, j)$, compute $\mathbb{P}(i, j)$ as the fraction of samples for which $X_\ell(i, j) = 1$.

We can obtain random (near-uniform) samples from $\mathcal{X}$ using either the Markov chain Monte Carlo (MCMC) method proposed by Gionis *et al.* [9] or the Sequential Importance Sampling (Sequential) algorithm proposed by Chen *et al.* [6]. MCMC guarantees uniformity of the samples, but it does not converge in polynomial time. Sequential produces near-uniform samples in polynomial time, but it requires $O(n^3 m)$ time per sample and thus using this algorithm to produce $N$ samples ($N >> n$) is beyond practical consideration. To recap, explicit sampling methods are impractical for large datasets; moreover, their accuracy depends on the number of samples $N$ and the size of the dataspace $\mathcal{X}$, which itself is hard to estimate.

## 4 Computing entry-wise PDFs

### 4.1 Warmup: The SimpleIS algorithm

With the aim to provide some intuition and insight, we start by presenting a simplified version of our algorithm called SimpleIS, also shown in Algorithm 1. SimpleIS computes the $\mathbb{P}$ matrix one column at a time, in arbitrary order. Each such computation consists of two steps: (a) *propose* and (b) *adjust*. The Propose step associates with every row $i$, weight $\mathbf{w}(i)$ that is proportional to the row marginal of row $i$. A naive way of assigning these weights is by setting $\mathbf{w}(i) = \frac{\mathbf{r}(i)}{m}$. We refer to these weights $\mathbf{w}$ as the *raw probabilities*. The Adjust step takes as input the column sum $x = \mathbf{c}(j)$ of the $j$th column and the raw probabilities $\mathbf{w}$ and adjusts these probabilities into the final probabilities $\mathbf{p}_x$ such that for column $j$

---

**Algorithm 1** The SimpleIS algorithm.

**Input:** $\langle \mathbf{r}, \mathbf{c} \rangle$
**Output:** Estimate of the PDF matrix $\mathbb{P}$
1: $\mathbf{w} = \text{Propose}(\mathbf{r})$
2: **for** $j = 1 \ldots m$ **do**
3:      $x = \mathbf{c}(j)$
4:      $\mathbf{p}_x = \text{Adjust}(\mathbf{w}, x)$
5:      $\mathbb{P}(:, j) = \mathbf{p}_x$

---

we have that $\sum_i \mathbf{p}_x(i) = x$. This adjustment is not a simple normalization, but it computes the final values of $\mathbf{p}_x(i)$ by *implicitly* considering all possible realizations of the $j$th column with column sum $x$ and computing the probability that the $i$th cell of that column is equal to 1.

Formally, if we use $\mathbf{x}$ to denote the binary vector that represents one realization of the $j$-th column of the hidden matrix, then $\mathbf{p}_x(i)$ is computed as:

$$\mathbf{p}_x(i) \coloneqq \Pr\left[\mathbf{x}(i) = 1 \mid \sum_{i=1}^{n} \mathbf{x}(i) = x\right]. \tag{2}$$

It can be shown [6] that Equation (2) can be evaluated in polynomial time as follows: for any vector $\mathbf{x}$, let $N = \{1, \ldots, n\}$, be the set of all possible positions of 1s within $\mathbf{x}$, and let $R(x, N)$ be the probability that exactly $x$ elements of $N$ are set to 1, i.e.,

$$R(x, N) \coloneqq \Pr\left[\sum_{i \in N} \mathbf{x}(i) = x\right].$$

Using this definition, $\mathbf{p}_x(i)$ is then derived as follows:

$$\mathbf{p}_x(i) = \frac{\mathbf{w}(i)R(x - 1, N \setminus \{i\})}{R(x, N)}. \tag{3}$$

The evaluation of all of the necessary terms $R(\ ,\ )$ can be accomplished by the following dynamic-programming recursion: for all $a \in \{1, \ldots x\}$, and for all $B$ and $i$ such that $|B| > a$ and $i \in B \subseteq N$:

$$R(a, B) = (1 - \mathbf{w}(i))R(a, B \setminus \{i\}) + \mathbf{w}(i)R(a - 1, B \setminus \{i\}).$$

**Running time:** The Propose step is linear in the number of the rows and the Adjust evaluates Equation (3) and thus needs at most $O(n^2)$ time. Thus, SimpleIS runs in time $O(mn^2)$.

**Discussion:** A natural question to consider is: why could the estimates of $\mathbb{P}$ produced by SimpleIS be inaccurate?

To answer this question, consider a hidden $5 \times 5$ binary table with $\mathbf{r} = (4,4,2,2,2)$ and $\mathbf{c} = (2,3,1,4,4)$ and assume that `SimpleIS` starts by computing the entry-wise PDFs of the first column. While evaluating Equation (2), `SimpleIS` generates all possible columns of matrices with row marginals $\mathbf{r}$ and a column with column sum 2 – ignoring the values of the rest of the column

| 0 | | | | | 4 |
|---|---|---|---|---|---|
| 0 | | | | | 4 |
| 0 | | | | | 2 |
| 1 | | | | | 2 |
| 1 | | | | | 2 |
| 2 | 3 | 1 | 4 | 4 | |

marginals. Thus, the realization of the first column shown in the matrix on the right is taken into consideration by `SimpleIS`, despite the fact that it cannot lead to a matrix that respects $\mathbf{r}$ and $\mathbf{c}$. This is because four more 1s need to be placed in the empty cells of the first two rows which in turn would lead to a violation of the column marginal of the third column. This situation occurs exactly because `SimpleIS` never considers the constraints imposed by the rest of the column marginals when aggregating the possible realizations of column $j$.

Ultimately, the `SimpleIS` algorithm results in estimates $\mathbf{p}_x(i)$ that reflect an entry $i$ in a column being equal to 1 conditioned over all matrices with row marginals $\mathbf{r}$ and a single column with column sum $x$ [6]. But this dataspace is not our target dataspace.

## 4.2 The `IS` algorithm

In the `IS` algorithm, we remedy this weakness of `SimpleIS` by taking into account the constraints imposed by *all* the column marginals when aggregating the realization of a particular column $j$. Referring again to the previous example, the input vectors $\mathbf{r}$ and $\mathbf{c}$ impose the following constraints on any matrix in $\mathcal{X}_{\langle \mathbf{r}, \mathbf{c} \rangle}$: column 1 must have least one 1 in the first two rows and (exactly) two 1s in the first five rows. These types of constraints, known as *knots*, are formally defined as follows.

**Definition 1.** *A knot is a subvector of a column characterized by three integer values $\langle [s, e] \mid b \rangle$, where $s$ and $e$ are the starting and ending indices defining the subvector, and $b$ is a lower bound on the number of 1s that must be placed in the first $e$ rows of the column.*

Interestingly, given $\langle \mathbf{r}, \mathbf{c} \rangle$, the knots of any column $j$ of the hidden matrix can be identified in linear time using an algorithm that recursively applies the Gale-Ryser condition on realizability of bipartite graphs. This method, and the notion of knots, were first introduced by Chen *et al.* [6].

At a high level, `IS` (Algorithm 2) identifies the knots within each column and uses them to achieve a better estimation of $\mathbb{P}$. Here, the process of obtaining the final probabilities is more complicated since it requires: $(a)$ identifying the knots of every column $j$ (line 3), $(b)$ computing the entry-wise PDFs for the entries in every knot (denoted by $\mathbf{q}_k$) (lines 4-7), and $(c)$ creating the $j$th column of $\mathbb{P}$ by putting the computed entry-wise PDFs back together (line 8). Note that we use $\mathbf{w}_k$ to refer to the vector of raw probabilities associated with cells in knot $k$. Also, vector $\mathbf{p}_{k,x}$ is used to store the adjusted probabilities of cells in knot $k$ given that $x$ 1s are placed within the knot.

---
**Algorithm 2** The `IS` algorithm.

**Input:** $\langle \mathbf{r}, \mathbf{c} \rangle$
**Output:** Estimate of the PDF matrix $\mathbb{P}$
1: $\mathbf{w} = \texttt{Propose}(\mathbf{r})$
2: **for** $j = 1 \ldots m$ **do**
3:      $\texttt{FindKnots}(j, \mathbf{r}, \mathbf{c})$
4:      **for** each knot $k \in \{1 \ldots l\}$ **do**
5:          **for** $x$: number of 1s in knot $k$ **do**
6:              $\mathbf{p}_{k,x} = \texttt{Adjust}(\mathbf{w}_k, x)$
7:          $\mathbf{q}_k = \mathbb{E}_x[\mathbf{p}_{k,x}]$
8:      $\mathbb{P}(:, j) = [\mathbf{q}_1; \ldots; \mathbf{q}_l]$

---

Step $(a)$ is described by Chen *et al.* [6], and step $(c)$ is straightforward, so we focus on $(b)$, which is the main part of `IS`. This step considers all the knots of the $j$th column sequentially. Assume that the $k$th knot of this column is given by the tuple $\langle [s_k, e_k] \mid b_k \rangle$. Let $x$ be the number of 1s inside this knot. If we know the value of $x$, then we can simply use the `Adjust` routine to adjust the raw probabilities $\mathbf{w}_k$. But since the value of $x$ may vary over different realizations of column $j$, we need to compute the probability distribution of different values of $x$. For this, we first observe that if we know that $y$ 1s have been placed prior to knot $k$, then we can compute lower and upper bounds on $x$ as:

$$L_{k|y} = \max\{0, b_k - y\}, \; U_{k|y} = \min\{e_k - s_k + 1, \mathbf{c}(j) - y\}.$$

Clearly, the number of 1s in the knot must be an integer value in the interval $[L_{k|y}, U_{k|y}]$. Lacking prior knowledge we assume that $x$ takes any value in the interval uniformly at random. Thus, the

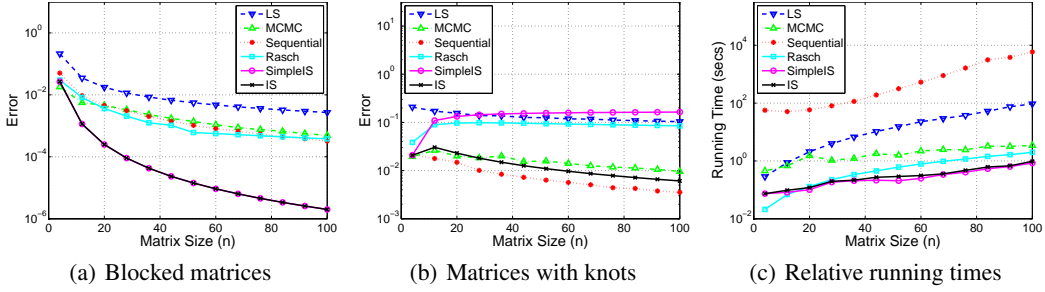

| (a) Blocked matrices | (b) Matrices with knots | (c) Relative running times |

Figure 1: Panels (a) and (b) depict *Error* (log scale) for six different algorithms, on two classes of matrices. Panel (c) depicts algorithmic running times.

probability of $x$ 1s occurring inside the knot, given the value of $y$ (i.e., 1s prior to the knot) is:

$$P_k(x|y) = \frac{1}{U_{k|y} - L_{k|y} + 1} \tag{4}$$

Based on this conditional probability we can write the probability of each value of $x$ as

$$P_k(x) = \sum_{y=0}^{\mathbf{c}(j)} Q_k(y) P_k(x|y), \tag{5}$$

in which $P_k(x|y)$ is computed by Equation (4) and $Q_k(y)$ refers to the probability of having $y$ 1s prior to knot $k$. In order to evaluate Equation (5), we need to compute the values of $Q_k(y)$. We observe that for every knot $k$ and for every value of $y$, $Q_k(y)$ can be computed by dynamic programming as:

$$Q_k(y) = \sum_{z=0}^{y} Q_{k-1}(z) P_{k-1}(y-z|z). \tag{6}$$

**Running time and speedups:** If there is a single knot in every column, `SimpleIS` and `IS` are identical. For a column $j$ with $\ell$ knots, `IS` requires $O(\ell^2 \mathbf{c}(j) + n\mathbf{c}(j)^2)$ – or worst-case $O(n^3)$ time. Thus, sequential implementation of `IS` has running time $O(n^3 m)$. This is the same as the time required by `Sequential` for generating a single sample from $\mathcal{X}_{\langle \mathbf{r}, \mathbf{c} \rangle}$, providing a clear indication of the computational speedups afforded by `IS` over sampling. Moreover, `IS` treats each column independently, and thus it is parallelizable. Finally, since the entry-wise PDFs for two columns with the same column marginals are identical, our method needs to only compute the PDFs of columns with *distinct* column marginals. Further speedups can be achieved for large datasets by binning columns with similar marginals into a bin with a representative column sum. When the columns in the same bin differ by at most $t$, we call this speedup $t$-`Binning`.

**Discussion:** We point out here that `IS` is highly motivated by `Sequential` – the most practical algorithm to date for sampling (almost) uniformly matrices from dataspace $\mathcal{X}_{\langle \mathbf{r}, \mathbf{c} \rangle}$. Although `Sequential` was designed for a different purpose, `IS` uses some of its building blocks (e.g., knots). However, the connection is high level and there is no clear quantification of the relationship between the values of $\mathbb{P}$ computed by `IS` and those produced by repeated sampling from $\mathcal{X}_{\langle \mathbf{r}, \mathbf{c} \rangle}$ using `Sequential`. While we study this relationship experimentally, we leave the formal investigation as an open problem.

## 5 Experiments

**Accuracy evaluation:** We measure the accuracy of different methods by comparing the estimates $\widehat{\mathbb{P}}$ they produce against the known *ground-truth* $\mathbb{P}$ and evaluating the *average absolute error* as:

$$Error(\widehat{\mathbb{P}}, \mathbb{P}) = \frac{\sum_{i,j} \left| \widehat{\mathbb{P}}(i,j) - \mathbb{P}(i,j) \right|}{mn} \tag{7}$$

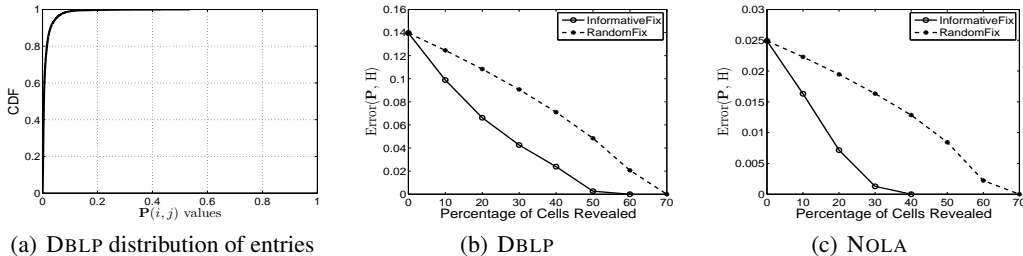

(a) DBLP distribution of entries          (b) DBLP           (c) NOLA

Figure 2: Panel (a) shows the CDF of estimated entry-wise PDFs by IS for the DBLP dataset. Panels (b) and (c) show $Error(\mathbb{P}, H)$ as a function of the percentage of "revealed" cells.

We compare the *Error* of our methods, i.e., SimpleIS and IS, to the *Error* of the two optimization methods: Rasch and LS described in Section 2.1 and the two explicit-sampling methods Sequential and MCMC described in Section 3. For MCMC, we use double the burn-out period (i.e., four times the number of ones in the table) suggested by Gionis *et al.* [9]. For both Sequential and MCMC, we vary the sample size and use up to 250 samples; for this number of samples these methods can take up to 2 hours to complete for $100 \times 100$ matrices. In fact, our experiments were ultimately restricted by the inability of other methods to handle larger matrices.

Since exhaustive enumeration is not an option, it is very hard to obtain ground-truth values of $\mathbb{P}$ for arbitrary matrices, so we focus on two specific types of matrices: blocked matrices and matrices with knots.

An $n \times n$ *blocked* matrix has marginals $\mathbf{r} = (1, n-1, \ldots, n-1)$ and $\mathbf{c} = (1, n-1, \ldots, n-1)$. Any table with these marginals either has a value of 1 in entry $(1, 1)$ or it has two distinct entries with value 1 in the first row and the first column (excluding cell $(1, 1)$). Also note that given a realization of the first row and the column, the rest of the table is fully determined. This implies that there are exactly $(2n-1)$ tables with such marginals and the entry-wise PDFs are: $\mathbb{P}(1, 1) = 1/(2n-1)$; for $i \neq 1$, $\mathbb{P}(1, i) = \mathbb{P}(i, 1) = (n-1)/(2n-1)$; and for $i \neq 1$ and $j \neq 1$, $\mathbb{P}(i, j) = 2n/(2n-1)$.

The *matrices with knots* are binary matrices that are generated by diagonally concatenating smaller matrices for which the ground-truth is computed known through exhaustive enumeration. The concatenation is done in such a way such that no new *switch boxes* are introduced (as defined in Section 3). While the details of the construction are omitted due to lack of space, the key characteristic of these matrices is that they have a large number of knots.

Figure 1(a) shows the *Error* (in log scale) of the different methods as a function of the matrix size $n$; SimpleIS and IS perform identically in terms of *Error* and are much better than other methods. Moreover, they become increasingly accurate as the size of the dataset increases, which means that our methods remain relatively accurate even for large matrices. In this experiment, Rasch appears to be the second-best method. However, as our next experiment indicates, the success of Rasch hinges on the fact that the marginals of this experiment do not introduce many knots.

The results on matrices with many knots are shown in Figure 1(b). Here, the relative performance of the different algorithms is different: SimpleIS is among the worst-performing algorithms, together with LS and Rasch, with an average *Error* of 0.1. On the other hand, IS with Sequential and MCMC are clearly the best-performing algorithms. This is mainly due to the fact that the matrices we create for this experiment have a lot of knots, and as SimpleIS, LS and Rasch are all knot-oblivious, they produce estimates with large errors. On the other hand, Sequential, MCMC and IS take knots into account, and therefore they perform much better than the rest.

Looking at the running times (Figure 1(c)), we observe that the running time of our methods is clearly better than the running time of all the other algorithms for larger values of $n$. For example, while both SimpleIS and IS compute $\mathbb{P}[]$ within a second, Rasch requires a couple of seconds and other methods need minutes or even up to hours to complete.

**Utilizing entry-wise PDFs:** Next, we move on to demonstrate the practical utility of entry-wise PDFs. For this experiment we use the following real-world datasets as hidden matrices.

DBLP: The rows of this hidden matrix correspond to authors and the columns correspond to conferences in DBLP. Entry $(i, j)$ has value 1 if author $i$ has a publication in conference $j$. This subset of DBLP, obtained by Hyvönen *et al.* [12], has size $17,702 \times 19$ and density $8.3\%$.

NOLA: This hidden matrix records the membership of $15,965$ Facebook users from New Orleans across 92 different groups [22]. The density of this 0–1 matrix is $1.1\%$[1].

We start with an experiment that addresses the following question: *"Can entry-wise PDFs help us identify the values of the cells of the hidden matrix?"* To quantify this, we first look at the distribution of values of entry-wise PDFs per dataset, shown in Figure 2(a) for the DBLP dataset (the distribution of entry-wise PDFs is similar for the NOLA dataset). The figure demonstrates that the overwhelming majority of the $\mathbb{P}(i, j)$ entries are small, smaller than 0.1.

We then address the question: *"Can entry-wise PDFs guide us towards effectively querying the hidden matrix $H$ so that its entries are more accurately identified?"* For this, we iteratively query entries of $H$. At each iteration, we query 10% of unknown cells and we compute the entry-wise PDFs $\mathbb{P}$ after having these entries fixed. Figures 2(b) and 2(c) show the $Error(\mathbb{P}, H)$ after each iteration for the DBLP and NOLA datasets; values of $Error(\mathbb{P}, H)$ close to 0 imply that our method could reconstruct $H$ almost exactly. The two lines in the plots correspond to RANDOMFIX and INFOR­MATIVEFIX strategies for selecting the queries at every step. The former picks 10% of unknown cells to query uniformly at random at every step. The latter selects 10% of cells with PDF values closest to 0.5 at every step. The results demonstrate that INFORMATIVEFIX is able to reconstruct the table with significantly fewer queries than RANDOMFIX. Interestingly, using INFORMATIVEFIX we can fully recover the hidden matrix of the NOLA dataset by just querying 30% of entries. Thus, the values of entry-wise PDFs can be used to guide adaptive exploration of the hidden datasets.

**Scalability:** In a final experiment, we explored the accuracy/speedup tradeoff obtained by $t$-Binning. For the the DBLP and NOLA datasets, we observed that by using $t = 1, 2, 4$ we reduced the number of columns (and thus the running time) by a factor of at least 2, 3 and 4 respectively. For the same datasets, we evaluate the accuracy of the $t$-Binning results by comparing the values of $\mathbb{P}_t$ computed for $t = \{1, 2, 4\}$ with the values of $\mathbb{P}_0$ (obtained by IS in the original dataset). In all cases, and for all values of $t$, we observe that the $Error(\mathbb{P}_t, \mathbb{P}_0)$ (defined in Equation (7)) are low – never exceeding 1.5%. Even the maximum entry-wise difference of $\mathbb{P}_0$ and $\mathbb{P}_t$ are consistently about 0.1 – note that such high error values only occur in one out of the millions of entries in $\mathbb{P}$.

Finally, we also experimented with an even larger dataset obtained through the Yahoo! Research Webscope program. This is a $140,000 \times 4252$ matrix of users and their participation in groups. For this dataset we observe that for 80% reduction to the number of columns of the dataset introduces an average error of size of only $1.7e^{-4}$ (for $t = 4$).

## 6 Conclusions

We started with a simple question: "Given the row and column marginals of a hidden binary matrix $H$, what can we infer about the matrix itself?" We demonstrated that existing optimization-based approaches for addressing this question fail to provide a detailed intuition about the possible values of particular cells of $H$. Then, we introduced the notion of entry-wise PDFs, which capture the probability that a particular cell of $H$ is equal to 1. From the technical point of view, we developed IS, a parallelizable algorithm that efficiently and accurately approximates the values of the entry-wise PDFs for all cells simultaneously. The key characteristic of IS is that it computes the entry-wise PDFs without generating any of the matrices in the dataspace defined by the input row and column marginals, and did so by implicitly sampling from the dataspace. Our experiments with synthetic and real data demonstrated the accuracy of IS on computing entry-wise PDFs as well as the practical utility of these PDFs towards better understanding of the hidden matrix.

**Acknowledgements**

This research was partially supported by NSF grants CNS-1017529, III-1218437 and a gift from Microsoft.

## Footnotes

[1]The dataset is available at: `http://socialnetworks.mpi-sws.org/data-wosn2009.html`

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
