[Reviews · NeurIPS 2013]

Submitted by Assigned_Reviewer_7

The paper gives a non-generative solution to the following problem: "given the row and column marginals of a hidden binary matrix H, what can we infer about the matrix itself?" Specifically, the paper advocates the computation of entry-wise PDFs of the hidden matrix via implicit sampling of the data space -- basically a method similar to EM, which the paper calls "propose" and "adjust".

The proposed approach is called "efficient" even though its computational cost is O(n^3) for each column!

The plots in Figure 1 need error bars.

The plots in Figure 2 are too small and hard to read.

In the "Utlizing entry-wise PDFs" Section, what was the AUC for "Can entry-wise PDFs help us identify the values of the cells of the hidden matrix?”

Typo: "whish"
Summary: This well-written paper addresses an interesting problem. A more thorough experimental section would have been nice.

An O(n^3) method is not efficient.

Submitted by Assigned_Reviewer_8

The paper provides a clear presentation of approaches available for
inferring properties of binary matrices when all that is known of them is
their row and column marginals. A new algorithm is presented that
provides probability estimates for each entry of the underyling binary matrix,
is parallelisable, and has overall running time that is equal to
the time used by sampling approaches to produce just a single sample.
Results are clearly presented and show the effectiveness of the results.

The paper was very well presented and clear and nicely described the
existing work available. Dealing with binary matrices is an important
topic regularly covered by papers at NIPS, and I think this paper
provides another interesting aspect of this problem type that many would
find interesting.

Most often it would be the case that the marginals are known
approximately (or are noisy) due to the nature of real binary matrices
(such as those from ratings obtained from human judgements),
which could results in a high-degree of inconsistency in the constraints
that they impose. Is there a way of providing this type of robustness.
Is t-binning one solution to handle such cases.

The paper also seems to have some interesting implications for privacy
preserving data analysis. Some comment on this might be interesting due
to the impact on what is shared about large datasets.
Summary: Overall, a well written paper, clear development of the algorithm and
context surrounding the work. The results show the effectiveness of the
method.

Submitted by Assigned_Reviewer_9

The paper concerns extracting reliable information about binary matrices, given only their column and row sums. These sums are considered as constraints on the space of possible matrices. The goal is to estimate the entry-wise probability density functions and to do this simultaneously via implicit sampling, making the proposed algorithm much more efficient than existing techniques. The paper is well written and easy to follow.

-- For sparse matrices, which is the case in the considered DBLP and NOLA datasets, the probability of having 1 is low, and the proposed approach computes low values as well. However, how useful is this information? It would be interesting to see how the algorithm performs on real or simulated matrices with density around 50%. Would we have many interpretable entries with probabilities close to 0 or close to 1, or would we get probabilities around 0.5?

-- Minor remarks:
1. Is there a reference for the considered LS approach with \bar h being the average entry value (Section 4.1)? If \bar h is close to 0.5, wouldn't this approach select an approximation with entries as close to 0.5 as possible, making the approximation less interpretable?
2. The legend of Figure 1(b) contains undefined labels.

-- Further questions:
1. Is the approach generalizable to integer matrices?
2. Is the proposed approach generalizable to the noisy case, i.e. how much would the entry-wise PDFs change if a given row or column sum is increased or decreased by one?
3. To speed up the proposed algorithm, columns having the same sum are considered to be the same and duplicates are removed from the computational step. Is it possible to extend the algorithm to be able to incorporate other types of prior knowledge? For example, a common assumption about data matrices is that they have low rank. (Redundant columns/rows lead to low rank matrices, but the low rank assumption is more general than that.)
Summary: The paper is interesting and clearly written. It would be interesting to see how the proposed algorithm behaves on 50% dense matrices and how it can be generalized to noisy data or in the presence of additional constraints, e.g., integer matrices or low-rank matrices.
Author Feedback

Author rebuttal: The reviewers provide a consistent and generally positive view of the paper, and challenge us with a set of well-motivated questions, such as: can we do well with noisy data, with integral data, and is a cubic running time really efficient? These thrusts are definitely deserving of further discussion and consideration in the future work section of our paper, but also highlight what we consider to be the most important point of our rebuttal: Work in this area, where the goal is to provide a model-free estimate of entry-wise PDFs of an unknown matrix based only on aggregate side-information, has broad applicability. Thus, a first paper defining the new notion of entry-wise PDFs as a data-analytic tool, and studying tractable methods for the important and natural case of binary data, has the potential for high impact. For example, as one of the reviewers points out, this understanding is a powerful tool as it enables data owners make important decisions as of whether to reveal the marginals for analysis.

Our responses to the reviewers’ comments follow:

On algorithmic efficiency:

Our algorithm addresses a problem for which an exact solution is predicating on counting of all binary matrices with fixed marginals, best methods for which take exponential time [23, 24]. Natural application of existing methods, such as explicit sampling, required at least O(n^5) time to be competitive. Therefore, achieving worst-case running time O(n^3) is efficient for this problem. Moreover, as discussed in the second-to-last paragraph of Section 4, O(n^3) is a loose upper-bound of the running time of our algorithm: the running time is more precisely a function of the number of knots within the column (k) and the column sum (c ) -- in which case the running time becomes O(k^2c+nc^2). In real datasets the number of knots rarely exceeds 3 or 4, and the column sum is small due to data sparsity. Thus, our algorithm is much faster in practice than its worst-case running time estimate suggests and readily enables off-line (“same day”) analysis for even very large datasets.


On noisy marginals:

Although we did not explicitly address the issue of noisy marginals, one can (and as one reviewer usefully suggested) view t-binning as adding noise to the marginals to improve efficiency. Adopting this viewpoint, our experiments reported in the last paragraph of Section 5 show that in most real datasets noisy marginals do not lead to significant errors in the entry-wise PDFs. We will clarify this important point in the next version.

On sparse and dense matrices:

Although real-world matrices are indeed sparse, our experiments with generated data focused on dense matrices. For these matrices (for which we could compute the ground truth entry-wise PDFs), the true PDF values are indeed close to 0 and to 1. For these matrices our algorithms had clearly the least error (see Figures 1(a) and 1(b)).

On generalizing our method to integer matrices:

For integer-valued matrices (or other discrete ranges), the entrywise PDFs will correspond to the distribution over the range of values allowed in every cell. Although some of the steps of the IS algorithm can be used for this case as well, it is not clear to us whether all the details can be worked out while retaining a workably efficient algorithm. We will pose this as an open question in the next version.


Some more specific responses to technical comments by the reviewers follow:

On SimpleIS vs. IS:

The main difference between SimpleIS and the IS algorithm is that the latter takes into account the knots, which as demonstrated by the results in Figure 1(b) -- where indeed we have a typo in the legend, “KnotIS” should be “IS” -- has a large benefit in matrices with many knots. Of course, we cannot rule out other weaknesses intrinsic to the IS algorithm. However, its design is based on solid principles, it is time-tested, and our experimental evaluation demonstrates that IS has the least error among all possible existing algorithms while being extremely efficient in practice.


On stability of the algorithm under row and column swaps:

The order of columns does not affect the result of the algorithm: the computations are independent for every column. Although the ordering of the rows also does not affect the result of the algorithm, in practice, we sort the rows by decreasing marginals to optimize the running time of the computation of the knots.

On using AUC as an evaluation measure:

The area under the curve (AUC) is not the right measure for our task because our goal is not to predict the actual value of 0 or 1 in the original dataset but -- as noted in Section 2.2, Equation (1) -- to estimate the entry-wise PDFs. Therefore, we evaluate our methods by comparing the estimated PDFs to the ground-truth PDFs.